# The Use of Metabolomics as a Tool to Compare the Regulatory Mechanisms in the Cecum, Ileum, and Jejunum in Healthy Rabbits and with Diarrhea

**DOI:** 10.3390/ani12182438

**Published:** 2022-09-15

**Authors:** Zheliang Liu, Jiahao Shao, Songjia Lai, Jie Wang, Kaisen Zhao, Tao Tang, Meigui Wang

**Affiliations:** 1College of Animal Science and Technology, Sichuan Agricultural University, Chengdu 611130, China; 2Farm Animal Genetic Resources Exploration and Innovation Key Laboratory of Sichuan Province, Chengdu Campus, Sichuan Agricultural University, Chengdu 611130, China

**Keywords:** intestinal barrier, metabolomics, antibiotic-free diet, intestinal inflammation, autophagy

## Abstract

**Simple Summary:**

The problems caused by antibiotic abuse have swept the world, and the Chinese government has responded to calls for a comprehensive ban on antibiotics. However, not using antibiotics also challenges China’s existing livestock industry. Based on this, we carried out a nontargeted metabolomics analysis of the jejunum, ileum, and cecum of diarrhea rabbits and normal rabbits fed with antibiotic-free diets, respectively, to find out the mechanism of action of each intestinal segment group and between different intestinal segments. The screened differential metabolites were mostly related to intestinal barrier, intestinal inflammation, and autophagy after a KEGG (Kyoto Encyclopedia of Genes and Genomes) analysis. In this paper, we analyzed the metabolic pathways that were significantly different between different intestinal segments and illustrated the mechanism and potential connections of the screened differential metabolites in different intestinal segments in the form of charts.

**Abstract:**

For many years, antibiotics in feed have been an effective and economical means to promote growth and disease resistance in livestock production. However, the rampant abuse of antibiotics has also brought very serious harm to human health and the environment. Therefore, the Chinese government promulgated laws and regulations on 1 July 2020, to prohibit the use of antibiotics in feed. To improve the effect of antibiotic-free feeding on China’s existing rabbit industry, we used the nontargeted metabolomics method to detect the differences between diarrhea rabbits (Dia) and normal rabbits (Con) on an antibiotic-free diet. A total of 1902 different metabolites were identified. A KEGG analysis showed that in the cecum, metabolites were mainly concentrated in bile secretion, antifolate resistance, aldosterone synthesis, and secretion pathways. The ileal metabolites were mainly concentrated in tyrosine metabolism, phenylalanine, tyrosine and tryptophan biosynthesis, steroid hormone biosynthesis, alanine, aspartate, and glutamate metabolism. The metabolites in the jejunum were mainly rich in panquinone and other terpenoid compound quinone biosynthesis, AMPK (adenosine 5′-monophosphate (AMP)-activated protein kinase) signal, inositol phosphate metabolism, and pentose phosphate pathway. After a deep excavation of the discovered differential metabolites and metabolic pathways with large differences between groups, it was found that these metabolic pathways mainly involved intestinal inflammation, intestinal barrier, and autophagy. The results showed that panquinone and other terpenoids could increase AMPK activity to promote cell metabolism and autophagy, thus trying to prevent inflammation and alleviate intestinal disease symptoms. In addition, we discussed the possible reasons for the changes in the levels of seven intestinal endogenous metabolites in rabbits in the diarrhea group. The possibility of improving diarrhea by adding amino acids to feed was discussed. In addition, the intermediate products produced by the pentose phosphate pathway and coenzyme Q had a positive effect on steroid hormone biosynthesis to combat intestinal inflammation.

## 1. Introduction

Animal husbandry is so dependent on antibiotics because antibiotics can prevent animal diseases [1,2], promote development, feed efficiency, delay digestion and absorption time, and improve animal appetite. However, due to imperfect regulatory systems and interest-driven factors, producers and farmers have abused or blindly added antibiotics, resulting in increased bacterial resistance and harm to human health [3], environmental ecology [4], and economic development. During the two years since the implementation of the ban in China, a large number of rabbits have suffered from gastrointestinal diseases due to multiple factors such as management and the environment [5]. Diarrhea is one of the clinical manifestations [6]. The gastrointestinal diseases of rabbits directly lead to a decrease in feed conversion rate, a decrease in production performance, and an increase in mortality, which cause great economic losses to the rabbit industry in China [7]. Since the current level of rabbit farming in China is difficult to rapidly improve, it is particularly important to reduce the negative impact of diarrhea on production in the current situation. The main causes of diarrhea in young rabbits in production practice include parasites [8], fungi [9], bacteria, and stress. Antibiotic-free feeding may lead to the structural imbalance of intestinal microflora and the destruction of intestinal epithelial integrity in young rabbits at the current level of culture in China, resulting in the increased invasion of pathogenic bacteria and increased inflammatory diseases. In order to inhibit the occurrence of inflammation and restore the integrity of the intestinal epithelial barrier, the body redistributes nutrients and energy [10], and the intestinal metabolic pathways become more active. The level of endogenous metabolites in tissues is also affected during the repair and energy redistribution of inflammatory tissues. For example, increased tryptophan metabolism inhibits inflammation and promotes intestinal barrier function.

In this study, we used metabolomics to determine the type and quantity of different metabolites in the intestine of diarrhea rabbits and normal rabbits. The differential metabolites were analyzed using the KEGG and the metabolic pathways with large differences between groups were screened out. The aim was to clarify the metabolites and metabolic pathways associated with intestinal inflammation and intestinal barrier function by analyzing and comparing the relationship between metabolic pathways in the same and different intestinal segments. It provides new ideas and a theoretical basis for the prevention and treatment of gastrointestinal diseases in rabbits in production practice.

## 2. Materials and Methods

### 2.1. Ethical Statements

This study was approved and implemented according to the ethical standards of the Animal Feeding and Use Committee of the School of Animal Science and Technology, Sichuan Agricultural University, Sichuan Province, 611130.

### 2.2. Animals, Feeding Management, and Sample Collection

This experiment took the Tianfu black rabbit as the experimental object. All the Tianfu black rabbits were raised on the Leibo rabbit farm. There were 105 breeding male rabbits and 768 breeding female rabbits at the Leibo rabbit farm. According to the conventional feeding and management methods at the rabbit farm, each rabbit was reared in a clean rabbit cage (600 × 600 × 500 mm) and was free to drink water. In January 2021, the rabbit farm began to use an antibiotic-free diet to feed rabbits in response to the ban. Subsequently, typical diarrhea symptoms such as deformity, watery stool, and sepsis occurred.

Six rabbits with typical diarrhea symptoms and six without diarrhea symptoms and similar body weight (845 g ± 20) were selected from 200 35-day-old female rabbits. The detailed procedure for collecting rabbit intestinal contents followed previous report by Tao et al. [11]. The jejunum, ileum, and cecum tissues were cut and washed with PBS, and then placed in an EP tube containing 4% paraformaldehyde. The contents of the jejunum, ileum, and cecum were collected immediately after slaughter and stored in 2 mL cryopreservation tube. Then, they were quickly frozen in liquid nitrogen and stored at −80 °C until sequencing analysis.

### 2.3. Morphological Analysis of Intestinal Tissue

Paraffin sections were dewaxed to water, hematoxylin staining nucleus, eosin staining cytoplasm, and dehydration sealing. The histopathological changes of the whole tissue biopsy were examined under a microscope (Olympus, Tokyo, Japan), and the normal microscopic imaging system area and obvious pathological changes were recorded.

### 2.4. UHPLC–MS/MS Analysis

We used a Vanquish ultrahigh-pressure liquid chromatography (UHPLC) system (German Waltham fisher company, Bremen, Germany) and Novogene Co., Ltd. (Beijing, China) Orbitrap Q Exactive TM HF mass spectrometer (fisher company, Bremen, Germany) for the UHPLC-MS/MS analysis. Gut samples’ injection was done using a Hypersil hypostyle column (100 × 2.1 mm, 1.9 microns), a linear gradient of 17 min, and a flow rate of 0.2 mL/min [12]. Orbitrap Q ExactiveTM HF mass spectrometer was set to work in the positive and negative polarities mode, the spray voltage was 3.2 kV, the capillary temperature was 320 °C, the sheath layer gas velocity was 40 arb, and the auxiliary gas flow velocity was 10 arb [13].

### 2.5. Data Processing and Metabolite Identification

We used compound found 3.1 (CD3.1, Thermo Fisher, Bremen, Germany) of the UHPLC–MS/MS system for processing the raw data files, for each metabolite in peak alignment, peak picking, and quantitatively. The main parameters were as follows: 0.2 m retention time tolerance; actual quality of tolerance, 5 PPM [14]; signal strength resistance, 30%; SNR 3; minimum strength, 100,000. Then, the returned signal was total spectral intensity peak intensity. According to the additive ions, the molecular ion peaks and fragment ions were determined, using a normalized data prediction formula. Then, we matched the peak against the MZ cloud (https://www.mzcloud.org/ (accessed on 9 November 2020)), mzVault, and MassList databases in order to obtain relatively accurate qualitative and quantitative results. We used statistical software tools R (R version rule 3.4.3) and Python (Python version 2.7.6) on CentOS (CentOS version 6.6). When chromatographic data did not have a normal distribution, the area normalization method was used to convert them to a normal distribution.

## 3. Results

### 3.1. Intestinal Pathological Features

The staining of intestinal tissues showed a coagulative necrosis of the jejunum mucosa in the Dia group and neat cells in the Con group. The number of lymphocytes in ileal lymph tissue in the Dia group decreased, and the number of lymphocytes in the Con group was normal. We observed a full-thickness coagulative necrosis of the cecum wall. On the contrary, the small intestine of the Con group was intact without pathological changes (Figure 1).

### 3.2. Metabolomics Changes in Diarrhea Rabbits

The correlation analysis of quality control samples showed that the correlation coefficient between the intestinal tissue metabolites data of quality control samples was close to 1. These results showed that the PLS-DA anion and cation model had a good prediction ability, and there was no overfitting (Figure 2).

### 3.3. Differential Metabolite Analysis

A differential metabolite analysis was used to identify differences between the Dia and Con groups of Hyplus rabbits, and the ileum, cecum, and jejunum metabolite screening criteria were: VIP score > 1, fold change > 1.5, or fold change < 0.667, and *p* values < 0.05. In total, 652 different kinds of metabolites were found in the appendix (485 kinds for the positive ion mode and 167 for the negative ion mode), 619 species in the ileum (399 in the positive ion mode and 220 in the negative ion mode), 631 species in the jejunum (positive ion mode, 459 species; negative ion mode, 172 species; Table 1).

The contents of 373 different metabolites in the cecum of the Dia rabbit group were higher than those of the Con group, while the contents of 278 different metabolites in the cecum of the Dia group were lower than those of the Con group. The contents of 411 different metabolites in the ileum were higher than those in Con rabbits and 225 metabolites were lower. Finally, compared with the control group, the contents of 424 different metabolites in the Dia rabbits’ jejunum were higher, while they were lower for 258 metabolites (Figure 3).

Cluster heat maps showing the distributions of different metabolites in the Dia and Con groups were similar, with trees indicating that the Dia and Con groups of rabbits’ ileum and cecum jejunum tissue samples could be separated (Figure 4).

### 3.4. KEGG Pathway Analysis

A total of 818 kinds of metabolites in the positive ion and negative ion modes were subjected to a KEGG pathway analysis, 297 of them for different metabolites (Table 2).

The metabolic analysis system (Figure 5) determined the differences in the enrichment of the metabolites of the metabolic pathways of key features. Seven endogenous metabolic pathways were significantly different between the DIA group and the control group, namely, the pentose phosphate pathway, AMPK signaling pathway, phosphoinositol metabolism, panquinone and other terpenoid-quinone biosynthesis, tyrosine, tryptophan, phenylalanine metabolism, and steroid hormone synthesis (*p* < 0.05).

## 4. Discussion

Diarrhea is one of the clinical manifestations of intestinal diseases in rabbits [15]. In this study, diarrhea rabbits and normal rabbits were collected as controls, aiming to preliminarily explore the mechanism of action related to diarrhea in rabbits under antifeeding conditions by histological and metabolomics methods. Histologically, the number of lymphocytes in most of the intestinal lymph follicles in the Dia group was significantly reduced. Pathological damage was mainly intestinal coagulation necrosis, followed by mucosal epithelial necrosis, shedding, erosion, and clinical diarrhea. On the contrary, the organs of rabbits in the Con group were normal. In terms of metabolomics, KEGG analysis of the screened differential metabolites showed that their metabolic pathways were mainly involved in intestinal barrier, oxidative stress, inflammatory immunity, and intestinal endogenous substance metabolism. We analyzed the most significant differential metabolic pathways and metabolites in each intestinal segment in order to find the potential relationship between the intestinal tract and the potential mechanism of action.

There were significant differences between groups in the jejunum anion mode, inositol phosphate metabolism, and pentose phosphate pathway. D-erythrose 4 phosphate, D-sedoheptulose 7 phosphate and glyceraldehyde 3 phosphate were detected in the pentose phosphate pathway, which were all upregulated compared with the control group. It indicated that the activity of the pentose phosphate pathway in diarrhea rabbits increased. One possible reason is that inflammatory mediators mediate metabolic disorders [16]. During infection, immune cells need to proliferate rapidly to inhibit the occurrence of inflammation. The demand of activated immune cells for glucose and ATP increases, which makes the glucose metabolism pathway and ATP generation pathway enhanced [17]. However, the specific mechanism is still unclear. Phosphoinositol metabolism and differential metabolites in phosphatidylinositol were also upregulated. The enhancement of the phosphatidylinositol metabolism in diarrhea rabbits may be caused by glucose metabolism disorder, and insulin is activated by plasma membrane measurement near phosphatidylinositol 3-kinases (PI3Ks) [18]. In addition to speculation, we also detected the PI3K-Akt signaling pathway in the ileum cation mode and cecum cation mode. The signaling pathway composed of PI3K and its downstream molecular signaling protein kinase B (Akt)/rapamycin target protein (mTOR) is one of the most important intracellular signaling pathways in mammals, which is essential for many important physiological functions, including cell cycle [19], cell survival, protein synthesis [20] and growth, metabolism, movement, and angiogenesis. In addition, the PI3K signaling pathway is also often associated with glucose metabolism [21], inflammation [22], immunity [23], and other diseases. This is in line with the results we found earlier.

In the jejunum cation mode, the AMPK signaling pathway and biosynthesis of panquinone and other terpenoid compounds such as quinone were significantly different between groups, and there was a regulatory role between them. AMPK is a key protein involved in multiple signaling pathways. Terpene compounds have the effect of activating the AMPK signaling pathway [24,25]. Once AMPK is activated, it can regulate almost all physiological and metabolic activities including life bodies. In recent years, the role of AMPK in regulating barrier function has also been gradually explored [26,27]. In this study, three differential metabolites were enriched in the AMPK signaling pathway and all the three differential metabolites were upregulated. Among them, berberine [28,29] and quercetin [30,31] have been confirmed by a large number of studies to have a positive effect on antioxidation, inhibiting inflammation and promoting intestinal barrier repair. However, the D-fructose 6 phosphate involved in the AMPK signaling pathway in this experiment has rarely been reported. We speculate that it may be due to the destruction of the intestinal barrier and the occurrence of inflammation, which lead to a spontaneous repair of the body and thus the redistribution of nutrients, such as the enhancement of the pentose phosphate pathway in the jejunum anion mode. Moreover, the PI3K-Akt signaling pathway and the mTOR signaling pathway were enriched in the ileum and cecum, and the intersection of these three pathways was autophagy. The decline of nicotinamide adenine dinucleotide (NAD) in cells is a feature of the decline of cellular immunity [32]. This study found that the levels of NAD + in each intestinal segment were downregulated, and an excessive downregulation of NAD + may lead to autophagy dysfunction through the combined action of the AMPK and mTOR signaling pathways [33]. The dysregulation of autophagy can lead to inflammation [34], autoimmune [35], or general immune diseases. Autophagy is a real immune process that permeates many aspects of innate and adaptive immunity. It has been reported that autophagy may indeed have evolved into one of the first antibacterial defenses available to eukaryotic cells. Autophagy has its own set of PRRs, the SLR adaptors, to eliminate them. Although the current research on immune autophagy is still in its infancy, autophagy is undoubtedly an opportunity target for the development of inflammatory diseases and new autoimmune therapies and may serve as an anti-infective mechanism.

In the ileum anion mode, compared with the control group, steroid hormone biosynthesis in diarrhea rabbits increased. We tried to find the potential relationship between the metabolites screened in this experiment and the increase of steroid hormone biosynthesis. We found that it may be affected by the pentose phosphate pathway [36] and coenzyme Q [37]. The pentose phosphate pathway can produce a large amount of NADPH to provide reducing agents for various synthesis reactions of cells, such as participating in the synthesis of fatty acids and sterols. More than 40 studies have shown that coenzyme Q has a positive effect on testosterone. However, there is little progress in related animal experiments.

Whether coenzyme Q can also regulate steroid hormones except testosterone remains to be studied. On the study of the inhibitory effect of steroid hormone synthesis on inflammation, Merk et al. reviewed the role of epithelial barrier as an alternative pathway for local and systemic synthesis [38]. Their roles in interorgan communication through correlated crosstalk were discussed to offset proinflammatory activities and prevent autoimmune activities. Bruscoli et al. found the positive effect of glucocorticoids in the treatment of inflammatory bowel disease and considered the corresponding anti-inflammatory properties of new glucocorticoid mediators [39]. Slominski et al. also found that deregulation of steroid and clavicle-forming signaling pathways could lead to various inflammatory and autoimmune diseases in a gender- and context-dependent manner [40]. The two steroid hormones androsterone and aldosterone screened in this study may be involved in the role of inflammation and immune regulation, which can provide some different prevention and treatment ideas for some inflammatory diseases and autoimmune diseases.

The repair of intestinal epithelial barrier damage caused by intestinal inflammation requires a lot of amino acid decomposition and synthesis [41]. A total of 14 amino acid metabolic changes were detected in the three intestinal segments of rabbits. The metabolic levels of three aromatic amino acids (tyrosine, tryptophan, phenylalanine) in different intestinal segments increased, and the difference of tryptophan metabolic levels was the largest. Numerous studies have shown that tryptophan metabolism is involved in intestinal immunity [42,43] and treatment [44]. The upregulation of methionine and cysteine metabolism detected in the ileum cation mode may be due to the fact that the main product of methionine and cysteine metabolism is glutathione [45]. Glutathione is a cellular antioxidant that can relieve intestinal oxidative stress [46] and alleviate inflammatory injury. Therefore, the metabolic level increased after intestinal inflammatory diseases. Moreover, methionine is also a methyl donor in many biochemical processes, generating more polyamines, thereby accelerating the proliferation of immune cells [45]. Compared with the normal group, the arginine and proline metabolism levels in the diarrhea group were upregulated. Arginine is an intestinal central metabolite. It is not only a component of protein synthesis, but also a regulatory molecule to maintain the physiological function of intestinal immunity [47], which plays an important role in intestinal immunity [48]. In mice, a diet supplemented with Arg is helpful to maintain intestinal barrier function [49]. Research on the relationship between proline and intestinal tract has gradually become a hot spot in recent years. Hu [50] et al. found that proline-rich small protein 2A (SPRR2A) was an intestinal antibacterial protein. It can selectively kill Gram positive bacteria by destroying the membrane. SPRR2A forms intestinal microflora, limiting bacterial association with intestinal surface and protecting bacteria from *Listeria* monocytogenes infection. This provides new insights and ideas for amino acids involved in immune response. In recent years, studies on the effects of amino acids on intestinal health and the function of mammals have explored the effects of glutamine, arginine, glutamic acid, tryptophan, methionine, and lysine [51,52,53]. However, there are few reports in rabbits. In addition to the above amino acids that have been proved to have effects on intestinal health, amino acids such as tyrosine and histidine, which are rarely studied, were also found in this study. Whether the relationship between amino acids in other species and intestinal health is applicable to rabbits and whether the selected tyrosine and histidine also affect the intestinal health of rabbits are our next research focus.

## 5. Conclusions

It was found that compared with the control group, the expression levels of the inositol phosphonic acid metabolism, phosphopentose pathway, steroid hormone synthesis, panquinone and other terpene quinone biosynthesis, and AMPK signaling pathway in the diarrhea group were increased. Intestinal inflammation led to a redistribution of energy. The metabolic level of damaged parts increased, indicating that the expression levels of the inositol phosphate metabolism and phosphopentose pathway were significantly upregulated. A large amount of NADPH generated by the enhancement of the pentose phosphate pathway provided a reducing agent for the synthesis of steroid hormones. Under the combined action of panquinone and NADPH, the synthesis of steroid hormones increased, thereby exercising its function of inhibiting inflammation. In addition, the difference between groups for the AMPK signaling pathway was large, and the detected levels of the three metabolites were upregulated. In this study, the synthesis of terpene compounds increased, thereby activating the AMPK signaling pathway. This study also found that the levels of NAD + in the three intestines decreased, and an excessive downregulation of NAD + could lead to the imbalance of autophagy. This may be due to the dysfunction of autophagy caused by the combined action of the AMPK and mTOR signaling pathways, thereby causing intestinal inflammation.

The metabolic levels of 14 amino acids were affected. In addition to confirming their relationship with intestinal health in other species, tyrosine and histidine were found in diarrhea rabbits for the first time. Whether these two will have an impact on the intestinal health of rabbits and the specific mechanism of action are our next focus of research.

## Figures and Tables

**Figure 1 animals-12-02438-f001:**
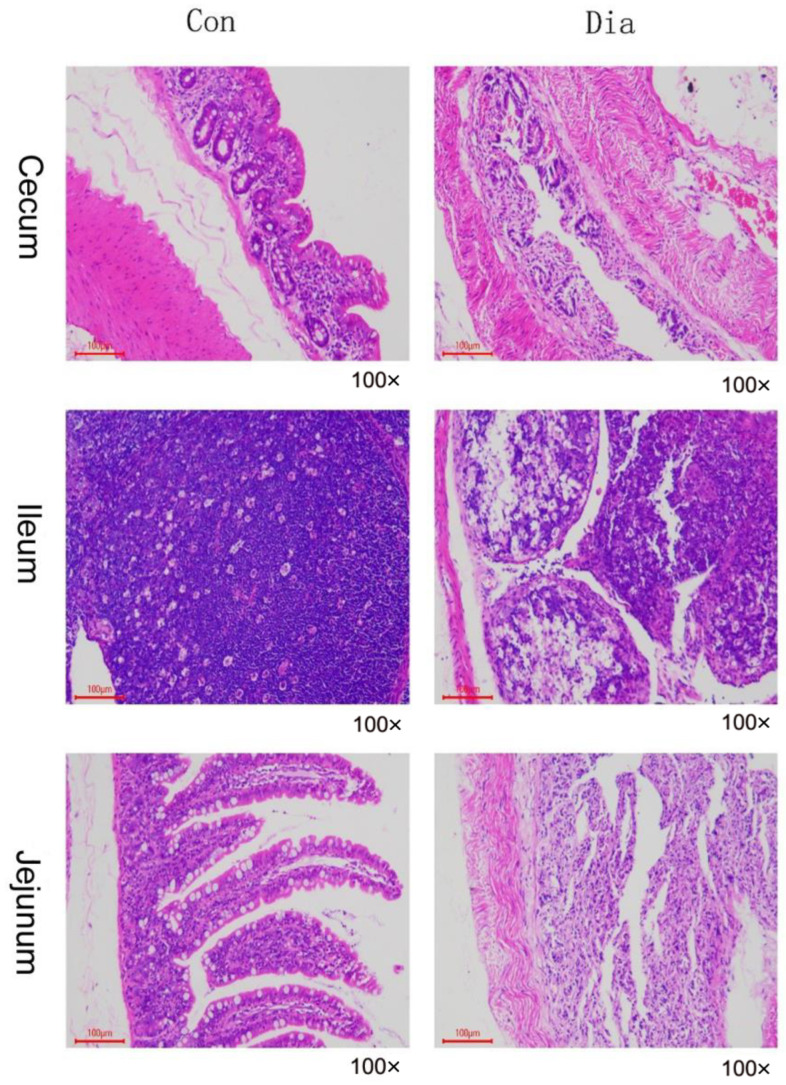
Normal rabbits’ (Con) and diarrhea rabbits’ (Dia) cecum, ileum, and jejunum tissue sections.

**Figure 2 animals-12-02438-f002:**
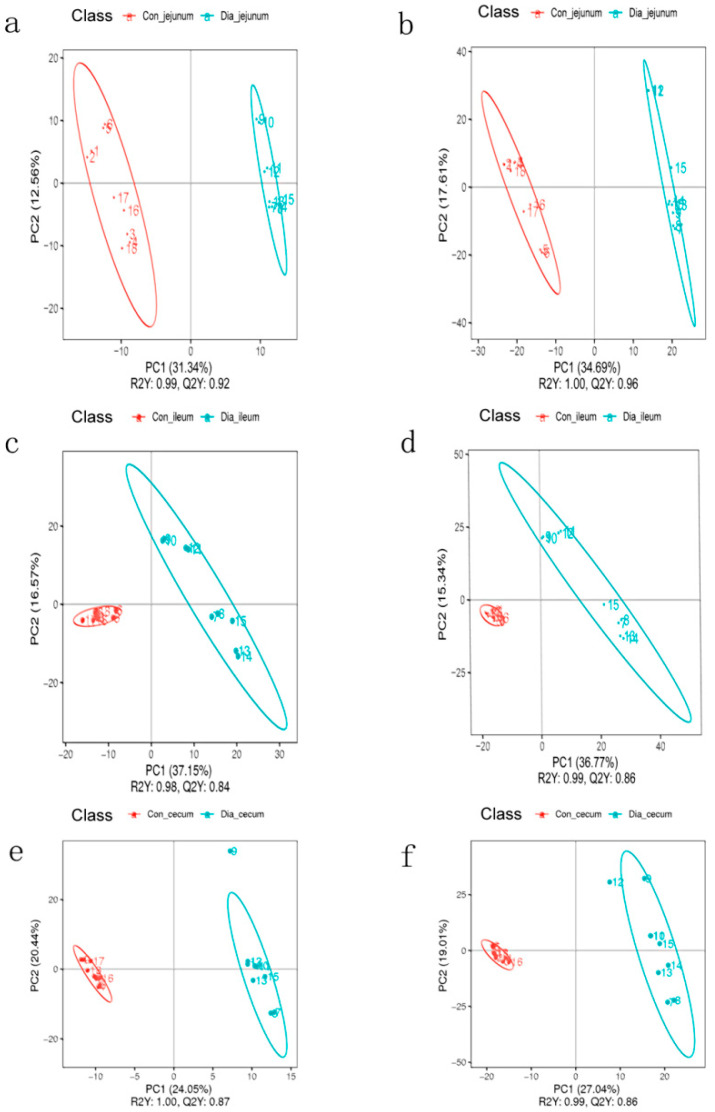
Partial least squares discriminant analysis (PLS−DA) was used to evaluate the sample differences between groups. (**a**,**b**) PLS−DA score map of jejunum in normal and diarrhea group under positive and negative ion modes; (**c**,**d**) PLS−DA score map of ileum in normal and diarrhea group under positive and negative ion modes; (**e**,**f**) PLS−DA score chart in positive and negative ion modes of cecum in normal and diarrhea groups.

**Figure 3 animals-12-02438-f003:**
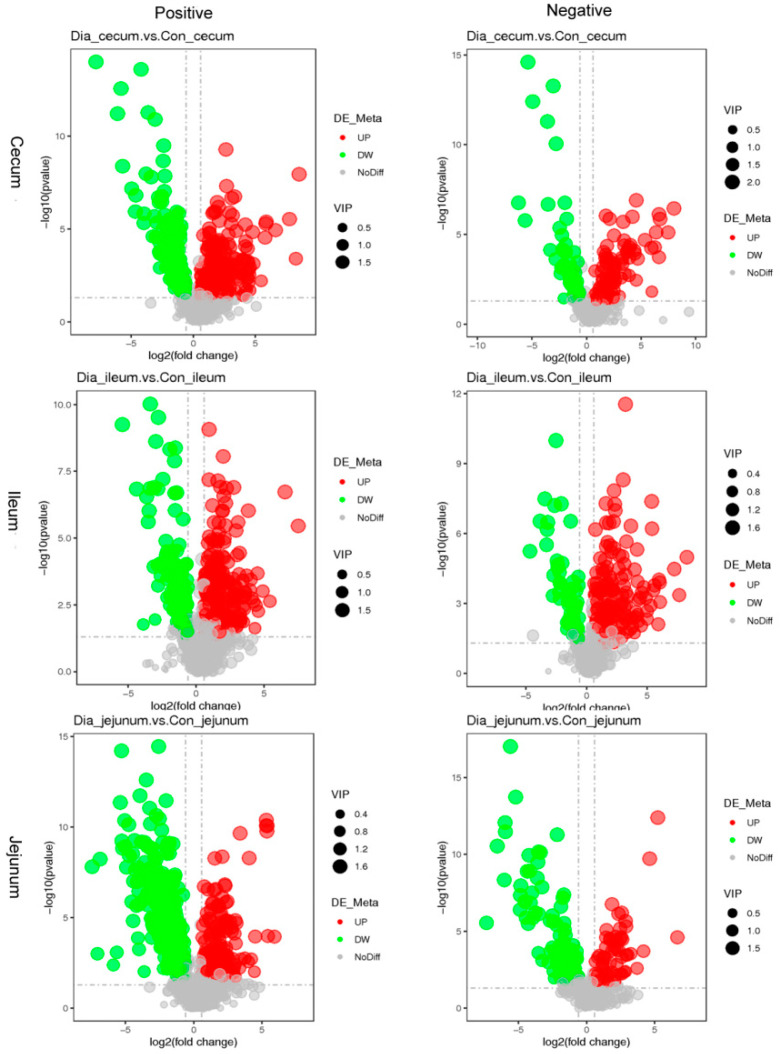
Volcano map of the difference in metabolites of Hyplus rabbits’ jejunum, ileum, and cecum tissue samples in the positive and negative ion modes. Green represents the downregulated metabolites, red represents the upregulated metabolites, respectively, different between groups, and gray represents the metabolites without significant changes. The abscissa indicated the change of differential metabolites between the Dia group and the Con group. The ordinate represents the significant level of differential metabolites between groups. Note: positive: cationic mode; negative: anion mode.

**Figure 4 animals-12-02438-f004:**
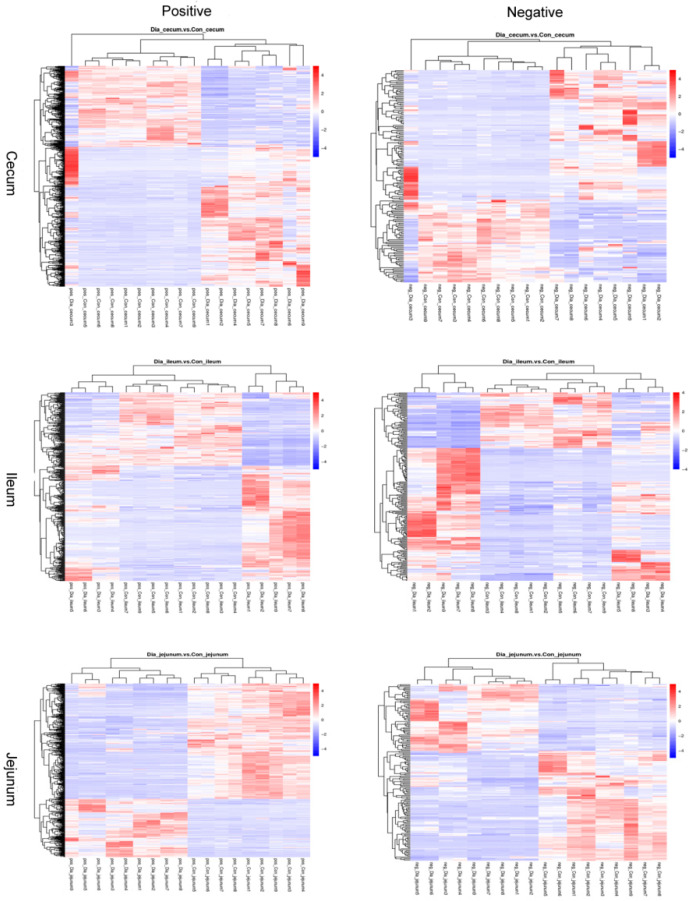
The difference in cluster heat maps of jejunum, ileum, and cecum tissue samples for the positive and negative ion modes.

**Figure 5 animals-12-02438-f005:**
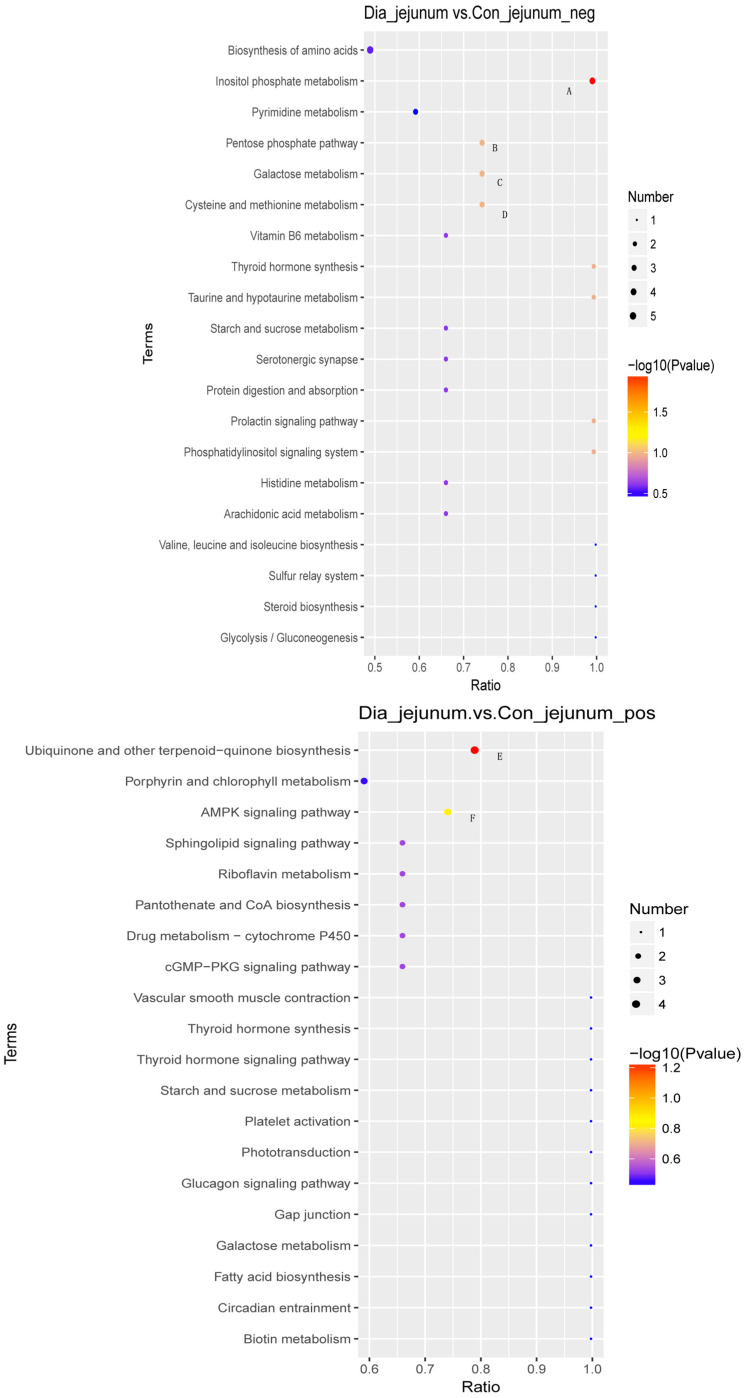
Bubble diagram for KEGG enrichment analysis of differential metabolites in positive and negative modes in jejunum, ileum, and cecum samples from Hyplus rabbits. A. Inositol phosphate metabolism; B. pentose phosphate pathway; C. galactose metabolism; D. cysteine and methionine metabolism; E. ubiquinone and other terpenoid−quinone biosynthesis; F. AMPK signaling pathway; G. phenylalanine, tyrosine, and tryptophan biosynthesis; H. steroid hormone biosynthesis; I. tyrosine metabolism; J. pentose phosphate pathway; K. alanine, aspartate, and glutamate metabolism; L. histidine metabolism; M. cysteine and methionine metabolism; N. arginine and proline metabolism.

**Table 1 animals-12-02438-t001:** In the cecum, ileum, and jejunum, the number of metabolites in diarrhea (Dia) group (not significantly different ^1^) was significantly higher than that in normal (Con) group (not significantly up ^2^). The number in the Dia Group was significantly lower than in the Con Group (not significantly down ^3^).

Intestinal Tissue Comparison	Ion Mode	Not Signif. Different ^1^	Not Signif. Up ^2^	Not Signif. Down ^3^
Dia_cecum vs. Con_cecum	Positive	485	307	178
	Negative	167	102	65
	Sum	652	409	243
Dia_ileum vs. Con_ileum	Positive	399	244	155
	Negative	220	155	65
	Sum	619	399	220
Dia_jejunum vs. Con_jejunum	Positive	459	154	305
	Negative	172	65	107
	Sum	631	219	412
All intestinal tissue comparisons	Total	1902	1027	875

**Table 2 animals-12-02438-t002:** Diarrhea group (Dia) and normal group (Con) Hyplus rabbits’ cecum, ileum, and jejunum tissue samples for KEGG enrichment analysis.

Tisue	Map ID	Map Title	*p*-Value	N	Meta IDs
Cecum	map04976	Bile secretion	0.073658105	105	Salicylic acid, Deoxycholic acid, Lithocholic acid, and Chenoaeoxychohc acid.
	map00520	Amino sugar and nucletide sugar metabolism	0.098581105	105	L-FucoseN-Acetylneuraminic acid, and N-Acetyl-alpha-D-glucosamine 1-phospnate.
	map00380	Trptophan metabolism	0.244398105	105	Picolinic acid, and Quinolinic acid.
	map01523	Antifolate resistance	0.072913182	182	Folic acid, Guanosine monophosphate and Adenosine 5′-monophosphate.
Ileum	map00140	Steroid hormone biosynthesis	0.144317115	115	Androsterone and Aldosterone.
	map00400	Phenylalanine, tyrosine, and tryptophan biosynthesis	0.155651115	115	D-Erythrose 4-phosphate, Phenylpyruvic acid, and 5-Phosphonbosyl 1-pyrophosphate
	map00250	Alanine, aspartate, and glutamate metabolism	0.050281	150	L-Glutamic acidzD-Glucosamine 6-phosphate and L-Asparagine.
	map00230	Purine metabolism	0.102723	150	2′-Deoxyadenosine, dAMP, Deoxyadenosine and Uric acid.
	map00430	Taurine and hypotaurine metabolism	0.137807	150	L-Glutamic acid and L-Glutamicacid.
Jejunum	map00562	Inositol phosphate metabolism	0.010489	103	D-Glucose 6-phosphate, D-myo-Inositol 1,4-bisphosphate, Inositol and Glyceraldehyde 3-phpsphate.
	map00030	Pentose phosphate pathway	0.103877	103	D-Erythrose 4-phosphate, D-Sedoheptulose 7-phosphate, Glyceraldehyde 3-phosphate.
	map00052	Galactose metabolism	0.103877	103	Dulcitol and Glyceraldehyde 3-phosphate.
	map00130	Ubiquinone and other terpenoid-quinone biosynthesis	0.058071	163	Menaquinone, Phylloquinone, gamma-Tocopherol and Shikonin.
	map04152	AMPK signaling pathway	0.135346	163	Berberine, Quercetin, and D-Fructose 6-phosphate.

## Data Availability

All data generated or analyzed during this study are included. All the figures and tables used to support the results of this study are included.

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
