# Peer review of "The Use of Metabolomics as a Tool to Compare the Regulatory Mechanisms in the Cecum, Ileum, and Jejunum in Healthy Rabbits and with Diarrhea"

_animals, 2022, doi:10.3390/ani12182438_

Round 1
Reviewer 1 Report
Dear Authors,
Very interesting research conducted on a reliable research group. They will probably be a valuable future guide for other researchers.
Minor revisions
Table 2 must be corrected because it is not readable.
Author Response
Dear Reviewer 3:
Thank you for your affirmation and approval of our research! Table 2 has been modified to read mode. If you have other questions, please feel free to contact us.

Reviewer 2 Report
The research material should be clearly described, as it is currently unclear whether the rabbits from both groups were fed a diet with antibiotics or without antibiotics:
Line 15-17: „Based on this, we carried out a nontargeted metabolomics analysis on the jejunum, ileum, and cecum of diarrhea rabbits and normal rabbits fed with antibiotics, respectively, to find out the mechanism of action of each intestinal segment group and between different intestinal segments”
Line 27-29: „we used the non-targeted metabolomics method to detect the differences between diarrhea rabbits (Dia) and normal rabbits (Con) in an antibiotic-free diet”
Line 90-95: „In January 2021, the rabbit farm began to use an antibiotic-free diet to feed rabbits in response to the prohibition of resistance. Subsequently, typical diarrhea symptoms such as deformity, watery stool, and sepsis occurred. Six rabbits with typical diarrhea symptoms and six without diarrhea symptoms and similar body weight (845 g ± 20) were selected”
Author Response
Comment 1: The research material should be clearly described, as it is currently unclear whether the rabbits from both groups were fed a diet with antibiotics or without antibiotics:
Response 1: I made a mistake in my writing. The rabbits from both groups were fed a diet without antibiotics.
(Line 15-17 in pages 1):Based on this, we carried out a non-targeted metabolomics analysis on of the jejunum, ileum, and cecum of diarrhea rabbits and normal rabbits fed antibiotics with antibiotic-free diets, respectively, to find out the mechanism of action of each intestinal segment group and between different intestinal segments.

Reviewer 3 Report
In this paper, authors analyzed the metabolic pathways of diarrhea rabbits and normal rabbits fed an antibiotic-free diet by means of non-targeted metabolomics method in order to detect possible differences. The approach to improve the animals’ feed to reduce the side effects of a diet without antibiotics is very interesting, also in light of the progressive worldwide ban of antibiotics in farming animals. However, there are some aspects that need to be addressed:
- Lines 14-15: “however…” what is the meaning of the sentence?
- Line 16: “analysis OF the jejunum..”
- Line 19: KEGG analysis…acronyms must be explained as they appear with the text...
- Authors should carefully revise the tense used throughout the text. For instance, the tense related to study results frequently changes between present and past....
- Line 39: AMPK, acronyms must be explained as they appear with the text...
- Line 49: what means save nutrition?
- Lines 62-65: Where are the citations about the first statement? It is true the opposite, in fact, it is known the AAD (antibiotic associated, diarrhea) as well as the dysbiosis are caused by antibiotics administration
- figure 4 should be improved because details cannot be observed
- what is figure 4/5? please clarify, moreover, improve the quality of the first two pictures…
- please also revise the references style
Author Response
Reviewer #2
Comment 1: Lines 14-15: “however…” what is the meaning of the sentence?
Response 1: Thanks very much for your suggestion. Actually,I made a mistake in my writing.Revised portions are highlighted in yellow.
(Line 14-15 in pages 1):However, without antibiotics ban also challenges China’s existing livestock industry. What I want to express is the impact on China's existing animal husbandry under the non resistant feeding environment.
Comment 2: Line 16: “analysis OF the jejunum..”
Response 2:Thanks very much for your suggestion. Revised portions are highlighted in yellow.
(Line 16 in pages 1):Based on this, we carried out a non-targeted metabolomics analysis on of the jejunum.
Comment 3: Line 19: KEGG analysis…acronyms must be explained as they appear with the text...
Response 3:Thanks very much for your suggestion.
(Line 19 in pages 1): KEGG(Kyoto Encyclopedia of Genes and Genomes) analysis.
Comment 4: Authors should carefully revise the tense used throughout the text. For instance, the tense related to study results frequently changes between present and past....
Response 4: Thanks very much for your suggestion. I have repeatedly checked the tense of the article and modified many errors.
Comment 5: Line 39: AMPK, acronyms must be explained as they appear with the text...
Response 5: Thanks very much for your suggestion.
(Line 39 in pages 1): AMPK(Adenosine 5‘-monophosphate (AMP)-activated protein kinase)signal.
Comment 6: Line 49: what means save nutrition?
Response 6: Thanks very much for your suggestion. Actually, what I want to express is the improvement of feed efficiency.
(Line 53 in pages 2):Animal husbandry is so dependent on antibiotics because antibiotics can prevent animal diseases, promote development, save nutrition feed efficiency.
Comment 6: figure 4 should be improved because details cannot be observed.
Response 6: Thanks very much for your suggestion.The details really can not be observed. My initial consideration was to avoid wasting a lot of space. Should I arrange each picture vertically in the article ?
Comment 6: what is figure 4/5? please clarify, moreover, improve the quality of the first two pictures…please also revise the references style.
Response 6: Thanks very much for your suggestion. I made a mistake in my writing.
(Line 210 in pages 11):Figure4,5 Figure 5. The quality of the first two pictures have improved and the references style also revised.
